# Program Translation via Code Distillation

**Yufan Huang[1,*]  Mengnan Qi[1,*]  Yongqiang Yao[1,*]  Maoquan Wang[1,*]**
**Bin Gu[2,3]  Colin Clement[1]  Neel Sundaresan[1]**
[1] Microsoft Cloud and AI
[2] School of Artificial Intelligence, Jilin University
[3] Mohamed bin Zayed University of Artificial Intelligence
`{yufanhuang,mengnanqi,yongqiangyao,maoquanwang}@microsoft.com`

## Abstract

Software version migration and program translation are an important and costly part of the lifecycle of large codebases. Traditional machine translation relies on parallel corpora for supervised translation, which is not feasible for program translation due to a dearth of aligned data. Recent unsupervised neural machine translation techniques have overcome data limitations by included techniques such as back translation and low level compiler intermediate representations (IR). These methods face significant challenges due to the noise in code snippet alignment and the diversity of IRs respectively. In this paper we propose a novel model called Code Distillation (CoDist) whereby we capture the semantic and structural equivalence of code in a language agnostic intermediate representation. Distilled code serves as a translation pivot for any programming language, leading by construction to parallel corpora which scale to all available source code by simply applying the distillation compiler. We demonstrate that our approach achieves state-of-the-art performance on CodeXGLUE and TransCoder GeeksForGeeks translation benchmarks, with an average absolute increase of 12.7% on the TransCoder GeeksforGeeks translation benchmark compare to TransCoder-ST.

## 1 Introduction

Program translation is the process of converting code written in one programming language to another. Unlike compilers that only convert high-level languages to low-level machine code, program translation can generate translations between complex high-level programming languages. The industry demand for high-quality program translation is ever-present, as many companies require the migration of project codes from older programming languages to new ones with better support, features, and a modern workforce. For example, COBOL

---

*Joint first author

code libraries that are widely used in banks and other financial institutions need to be migrated to updated languages for easier maintenance and development. Additionally, adaptation to different hardware platforms and operating systems often requires projects to be implemented in different high-level languages or different versions of the same language. An automatic program translation system, such as Java to Swift, is a blessing for Android developers who want to adapt their applications to the iOS operating system.

Natural language translation typically relies heavily on high-quality parallel corpora. However, creating parallel data for program translation tasks can be prohibitively expensive as it requires the generated code to be as aligned as possible. Unlike natural language, even minor variations or missing tokens can result in significant errors in code languages. Additionally, perfect alignment of tokens does not guarantee accuracy due to unique syntax rules and restrictions present in each code language. For example, when translating Java code to C++, the variable name "xor" in Java cannot be directly mapped as a variable name in C++ because it is a reserved word in C++. Therefore, program translation is a challenging task due to the difficulty in obtaining high-quality parallel code data.

Recent research has proposed unsupervised methods for neural machine translation in program translation tasks. PLBART (Ahmad et al., 2021) randomly adds various types of noise (token deletion, token shuffling, etc.) to multilingual code snippets and forces the model to recover them. This strategy not only helps close the gap between different programming languages but also improves the alignment ability of the model. TransCoder (Roziere et al., 2020), introduced an unsupervised approach for translating source code by constructing a pseudo-parallel corpus using back-translation to augment translation data pairs. Another system, TransCoder-ST (Roziere et al.,

2021b), employs open-source automatic test generation tools to develop unit tests for commonly used programming languages. This system eliminates potentially invalid data pairs in back-translation and refines the unsupervised model using high-confidence data examples that have passed the tests, thus enhancing the overall level of confidence in the machine translation process.

These methods, however, face significant challenges, which currently limit their ability to serve as a perfect method of translation. Due to the random noise, PLBART may end up paying more attention to irrelevant information in the input, overlooking the crucial information needed for alignment, resulting in slower convergence speed and reduced performance. TransCoder models can be biased towards certain data patterns, which can result in poor performance on new or unseen data. Besides, the implementation of back translation in language alignment can introduce randomness and ambiguities between program languages, which ultimately slows the convergence of the model and presents challenges when aligning low-frequency words. To alleviate this problem, TransCoder-ST filter out these errors during back translation though unit tests, but the ensuing challenge is the cost of constructing unit tests, which can be particularly high in industrial projects where the code requires multiple dependencies. This poses a significant obstacle in the capturing of appropriate test data for code fragments in complex scenarios and at times, make it infeasible.

In this paper, we introduce code distillation, which overcomes the limitations of parallel data and alignment in different languages by leveraging a custom compiler to a simplified intermediate representation which contains only the essential logical aspects and object names. We train a model to decompile the distilled code, so that the process of translation is to compile the source snippet to the distilled representation, and decompile to the target language using the trained model. We systematically discovered which elements of each source language are necessary for code understanding, eliminating these language-specific elements from the distilled code representation. Further, our distillation unifies basic morphemes, blurring grammatical differences between languages, and uses a more abstract form to merge high-level semantic expressions between different languages. To demonstrate the effectiveness of using distilled code as

translation pivot, we introduce a novel multilingual program translation model called CoDist that unifies multiple translation pairs into a single distilled code decompiler task. Through the specific language token switch, the distilled code derived from a source language code snippet can be seamlessly converted into any target language supported by the model. Our experiments demonstrate that our method surpasses state-of-the-art techniques on the CodeXGLUE (Lu et al., 2021) and TransCoder GeeksforGeeks benchmark datasets. Furthermore, our model exhibits zero-shot and few-shot abilities that are competitive in low-resource code languages. To summarize, our main contributions are:

1. In this study, we introduce a pivot language in programming translation called distilled code. The code distillation technique is a lossy compression, extracting the core semantics of any code snippet of high-level programming languages into a language agnostic format. Distilled code unifies all basic morphemes across languages. For high-level semantics (e.g., complex API calls), code distillation extracts the core part of program logic, and decomposes concatenated object names into bag-of-words representations.

2. Through our novel multilingual program translation model, we unify multiple translation pairs into a single distilled code decompiler task. To build a code translation system that support $N$ programming languages translation, our system only need to train one neural network model instead of $N^2$ sequence-to-sequence model tasks by traditional translation techniques.

3. Our method can leverage massive unsupervised code corpora through self-supervised pre-training tasks, and our final pre-training form is identical to the program translation form, which reduces the gap between pre-training and fine-tuning. It is noteworthy in that our training does not rely on parallel corpora, and can produce translations of much higher quality than traditional methods.

4. Our method outperforms current state-of-the-art techniques on the CodeXGLUE and TransCoder GeeksforGeeks benchmark datasets, and it is competitive in low-resource code languages.

## 2 Related Work

The lack of parallel translation pairs poses a significant challenge for automated program translation. To overcome this challenge, many recent works have focused on designing pre-training tasks or using various unsupervised methods. In this section, we will discuss the benefits and limitations of these approaches.

### 2.1 Pre-training for Program Translation

The goal of the paradigm of large-scale pre-training is to stimulate the potential of models in program translation by designing well-structured pre-training tasks. CodeBERT (Feng et al., 2020) pre-trains the masked language modeling task proposed by BERT (Devlin et al., 2018) on code corpora, then adds a decoder for end-to-end training on program translation. However, it treats the code as ordinary text and loses the structural information of the code. Some works incorporate intrinsic features of programming languages to overcome this shortcomings. GraphCodeBERT (Guo et al., 2020) improves on CodeBERT (Feng et al., 2020), adding data flow graphs extracted from source code to improve the model's understanding of code structure. StructCoder (Tipirneni et al., 2022) improves the transformer model to make encoder decoders structure-aware, and introduces abstract syntax tree and data flow graph information. Most of these models are encoder-only pre-trained models that focus on code understanding, the lack of pre-training in the decoder consistently restricts the model's performance on the task of program translation.

Some researchers begun to investigate the joint pre-training of the encoder and decoder. PLBART (Ahmad et al., 2021) builds on the existing natural language translation model BART (Lewis et al., 2019), continuing the same pre-training on code domain data. They incorporate various forms of noise, including token masking, deletion, and infilling, into the input code and force the model to reconstruct the original code at the decoder. MuST-PT (Zhu et al., 2022b) pre-trained on multilingual code snippets for translation task. Those pre-training tasks is still disparate from program translation task, making them still rely on fine-tuning using parallel program translation data sets.

### 2.2 Unsupervised for Program Translation

Unsupervised program translation aims to automatically translate code from one programming language to another without the need for parallel data or human supervision. TransCoder (Roziere et al., 2020) combines cross-lingual masked language modeling (Lample and Conneau, 2019), denoising auto-encoding, and back-translation (Artetxe et al., 2017) and applies them to the source code setting. However, back translation creates many pseudo-parallel code pairs with a lot of noise, which restricts the upper limit of the model's ability. TransCoder-ST (Roziere et al., 2021b) adds an automated unit-testing system to get high-quality pseudo-translation pairs in back translation. After filtering by unit test, the model can align on some common basic elements, but it still cannot generate correctly in complex scenarios such as industrial projects where the code requires multiple dependencies.

## 3 Distilled Code Compiler

Since these approaches mentioned in the "Related Work" section still have their own limitations, we choose an alternative approach that involves building a translation pivot. Instead of one-to-one translation processes, we first compile any source language program to a distilled code representation and then decompile it into target program using a trained model. We provide a unified front-end for compilation, which improves information density and narrows the language gap by distilling core semantics from the original code. In this section, we will briefly introduce translation pivots and explain the construction process of our compiler.

### 3.1 Brief Review of Translation Pivot

TransCoder-IR (Szafraniec et al., 2022) explored a new path for program translation, it utilizes a low-level intermediate representation (IR), provided by LLVM (Lattner and Adve, 2004), to be the pivot between the source code and the target code. LLVM IR is a low-level programming language used as an intermediate step in the compilation process between a high-level programming language and machine code, it provides a relatively uniform representation for different programming languages. In their experiment, the source code is converted into LLVM IR through the compiler, and LLVM IR is restored into the target code. Its great advantage is that, the process of translating source language code to the LLVM IR does not require additional neural network training, and training a neural decompiler should suffice to obtain the tar-

get language code.

According to (Szafraniec et al., 2022), the experimental results were limited by the huge variations between different IR dialects in LLVM. The LLVM ecosystem is jointly developed by many teams, which leads to different IRs for each language. On the other hand, the original intention of LLVM IR design is not just to align different languages, so there is a lot of redundant information that has weak correlation with program translation (such as for garbage collection and debugging). These factors lead to LLVM IR to be extremely verbose, so that even short programs can consume the entire limited context window of LLMs.

## 3.2 Our Distilled Code Compiler

A well-designed translation pivot should be a independent from specific languages and still be able to capture the semantics of the input as much as possible. We design a new set of compilers called distilled code compilers for languages such as Python, C++, Java and C#. Compare with the intermediate representations from the low-level compiler like LLVM, our distilled code contains less information from the source language, while preserving overall logical flow and data type relations, which makes it more consistent with the design principles of a translation pivot.

Specifically, our distilled code compiler first convert origin codes into abstract syntax tree, and then unimportant language-specific elements are removed from the tree and reassembled into code forms that embody the original logic and data flow. Figure 1 depicts the comprehensive procedure of code distillation.

### 3.2.1 Distillation

The abstract syntax tree (AST) serves as a graphical representation of the grammatical structure of a programming language, which is a natural medium for expressing distillation operations on different code parts. We compute the AST by using a parser generator tool called TreeSitter[1]. Code distillation compresses language-specific information and reduces variations in programming languages while preserving the input code semantics to the greatest extent possible. This section will describe the distillation process in details.

**Syntax Tree Pruning** The experiments documented in Appendix A reveal the impact of various code components on the semantic representations of the data. It is concluded, based on the findings in Appendix A, that moderate elimination of non-central information will not significantly affect the semantic understanding of the input code by pre-trained models such as PLBART. As a result, we have implemented a pruning technique to filter out non-significant syntax tree nodes while retaining crucial components such as variable names, function names, data types, and reserved keywords. Despite some loss of information during the pruning process, the utilization of a robust language model enables us to restore it in the decompilation stage.

**Unify Basic Morphemes** While the logical flow of the code components are retained in pruned syntax trees are preserved, various high-level programming languages own a distinct set of symbols and keywords to express semantics. In Appendix B, the design scheme for mapping the basic morphemes in Python, Java, C++, and C# languages into a unified expression form is presented and discussed in detail. Additionally, the existing program translation model faces obstacles related to the unique expression form of each language. A case in point is the switch expression in Java, which is not supported by the current Python and must be substituted with an if-else structure to achieve the same result. We also unify these specialized expressions from various languages into a unified form.

**Fuzzing Remaining Variations** Different languages often have library and API names which have similar sub-word elements but differ in subtle ways like sub-word order or choice. We overcome this difference by representing function and object names by a bag of words representation of the snake and camel-case segmented elements. Further, we randomize the order of these during decompiler training to ensure the model is not sensitive to language-specific orderings except in the decompilation stage.

### 3.2.2 Reassembling

The final step involves reassembling the syntax tree into a sequence format that is acceptable for use with the language model. Special string templates are designed for morphemes shared by all programming languages such as for loops, if statements, and while loops. These templates take into account the design patterns of different languages, retaining causal, nested, referring and other logical relationships between different components in the syntax tree, resulting in a comprehensive, expressive, and

---

[1] https://tree-sitter.github.io/tree-sitter/

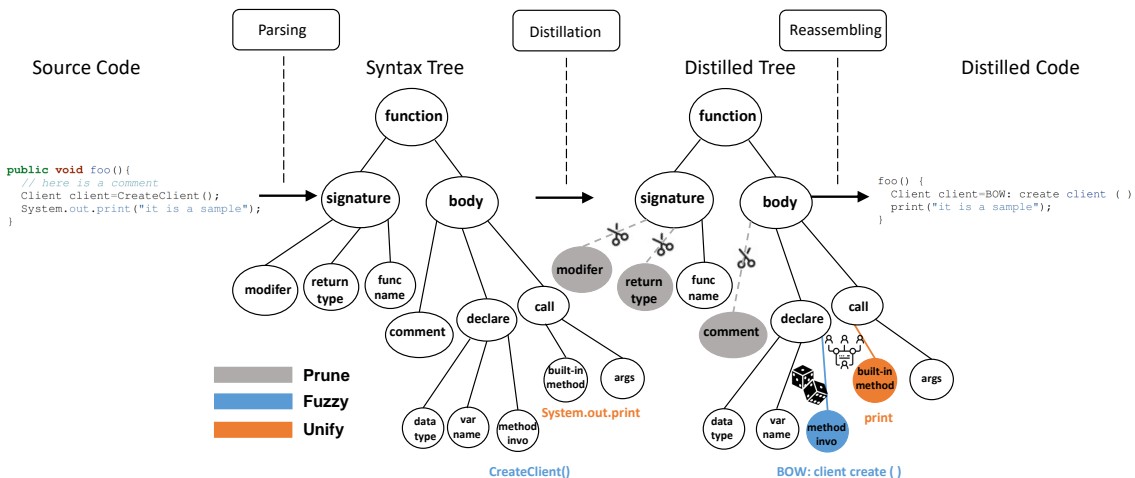

Figure 1: **The overall pipeline for converting source code into distilled code.** In the first step, the source code is represented as a syntax tree. Next, we align different languages to translation pivots by removing language-specific elements which are not important to the logical and semantic function. Finally, the nodes are combined in string templates to yield distilled code strings suitable for large language model training.

extensible representation.

## 4 Distilled Code Decompiler

After source code is transformed into the translation pivot through the distilled code compiler, next we need to build a decompiler that can output the target code. We introduce our multilingual distilled code decompiler model, which is a encoder-decoder transformer based model pre-trained on large-scale multilingual code corpora. Under the framework of multilingual models, all languages share the same encoder and decoder and map to the same latent space. In this section, we introduce several self-supervised pre-training tasks that we implemented. The first two tasks provide good representations for our distilled code and original code. The last pre-training task is similar to the final decompiler task, which further aligns the distilled code and target code.

### 4.1 Problem Setting

In this work, the primary focus will be on program translation at the function level, as these sequences embody a minimal semantic program of an appropriate length. Suppose our model supports the following set of code languages $\mathcal{C} = \{\mathcal{C}_1, \ldots, \mathcal{C}_k\}$, where each $\mathcal{C}_i$ is the monolingual code corpora in language $i$. Giving a source code function $\mathbf{X} = \{x_1, x_2, \ldots, x_l\}$ in code language $\mathcal{C}_i$, which consists of $l$ code snippets. It is first input to the distilled code compiler to obtain its corresponding distilled code $\mathbf{D^x} = \{d_1^{(\mathbf{x})}, d_2^{(\mathbf{x})}, \ldots, d_l^{(\mathbf{x})}\}$, the goal

of program translation is to generate a target code function $\mathbf{Y}$ in code language $\mathcal{C}_j$.

### 4.2 Representation Pre-training

There are a substantial amount of high-quality code corpora available for different program languages on internet. However, these monolingual codes are not aligned across programming languages, which presents a significant obstacle for conventional supervised training methods to effectively utilize them for translation. To overcome this challenge, we propose a set of unsupervised pre-training methods that aim to mine the hidden semantic information in these data. The pre-training methods discussed in this subsection allow the model to obtain a more robust embedding for translation tasks. **Masked Language Modeling** First introduced in (Devlin et al., 2018), Masked Language Model (MLM) pre-training task is a self-supervised learning task designed to train models to understand context and semantics in natural language. In this work, we extend the application of the aforementioned method to our translation pivot, the distilled code. We propose to train our model simultaneously on multiple programming languages, leveraging the distilled code as a unified representation. This approach has the potential to improve the performance of our model on cross-lingual programming tasks. Specifically, we first obtain the distilled codes for the multilingual program language codes through the distilled code compiler, concatenate it with source code and then train the encoder to predict randomly masked tokens.

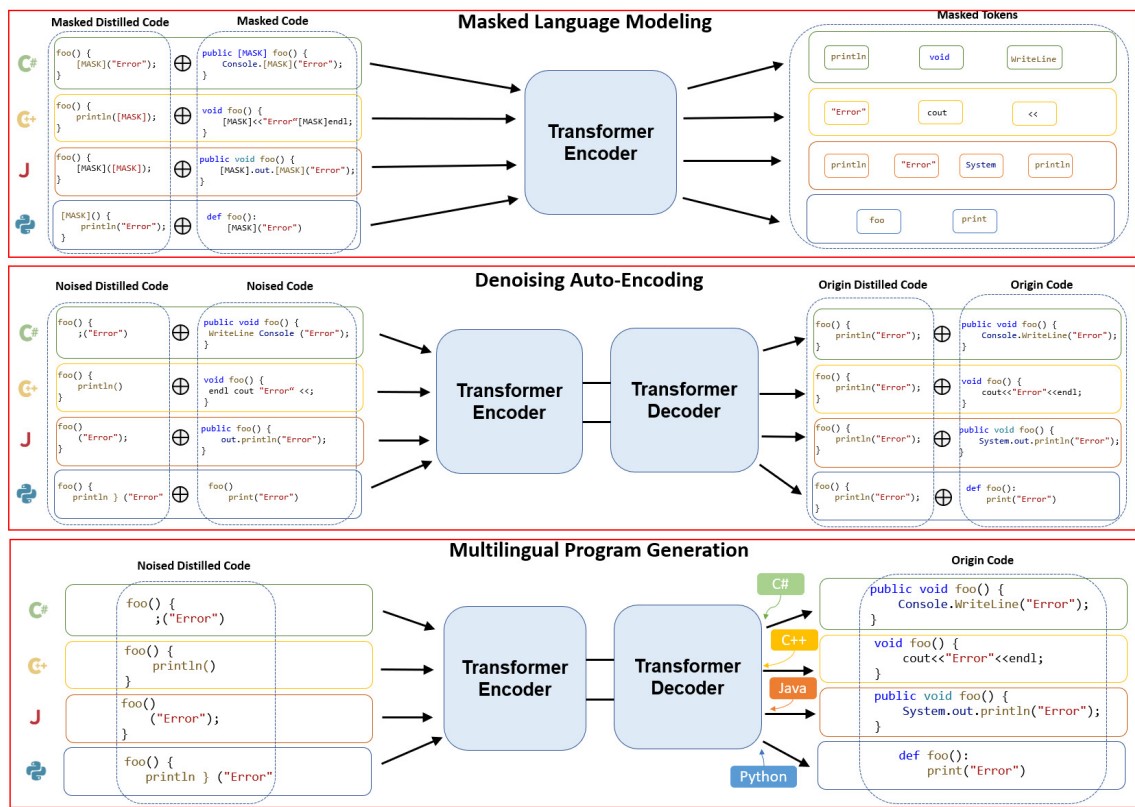

Figure 2: **The whole process of pre-training.** From top to bottom, we implemented three pre-training tasks step by step. Our pre-training first allows the model to fully understand the domain data, and then moves closer to the downstream task decompiler form.

**Denoising Auto-Encoding** Inspired by the BART model, our approach incorporates various types of program corruption on both the word and sentence level, such as random word shuffle, random word dropout, and sentence permutation. Both the source code and the corresponding distilled code are intentionally corrupted, and then combined into a single sequence. The model is then trained to reconstruct the original sequence. To enhance the robustness of the model decompiler, we specifically add more noise to the code snippets stored in the bag of words in the distilled code.

### 4.3 Training for Translation

Upon completion of the pre-training phase described in the previous subsection, our model has already obtained a good embedding representation for distilled codes from different program languages. The pre-training task outlined in this section further assists the model in aligning distilled code and various target codes.

**Multilingual Program Generation** We start from the distilled code to generate the complete code across various programming languages. During the pre-training stage, our model utilizes programming language code and its corresponding distilled code in each monolingual corpus as a training data pair. In order to aid the model in accurately translating to the corresponding language, special language symbols (e.g., <java>, <python>) are added as a switch on the decoder. There are subtle differences between translation pivots distilled from different languages. To narrow the gap among them, we add noise during the training process. After training, the model is able to uniformly understand distilled codes in different languages and generate corresponding target codes on the decoder side.

## 5 Experiments

### 5.1 Training Details

**Training Datasets** We select C#, C++, Java, and Python files from projects with more than 5 stars in the GitHub public repositories and CodeNet projects (Puri et al., 2021). We extract complete, structurally sound code at the function level and preprocess the data by deleting docstrings, comments, and dead code blocks. We also employed XLCoST (Zhu et al., 2022a), a benchmark dataset

that contains fine-grained parallel data in seven commonly used programming languages.

**Evaluation** Studies focusing on code translation typically use the CodeXGLUE dataset (Lu et al., 2021) (Java to C#, C# to Java) as the evaluation data. We follow them and utilize the two metrics offered by the benchmark, BLEU (Papineni et al., 2002) and CodeBLEU (Ren et al., 2020), to assess the performance of our model. These metrics can measure n-gram overlap, code syntax, and semantic equivalence between the generated code and the target code. Compared with these metrics that reflect the quality of the generated code, sometimes humans expect to intuitively feel whether the translation result can run correctly and return the same result as the ground truth. GeeksforGeeks is an online platform with computer science and programming articles, which gathers many coding problems and presents solutions in several programming languages. The TransCoder model provides test environments and test samples for the C++-Python-Java code pairs collected from GeeksforGeeks. They evaluate whether the generated function returns the same output as the reference when given the same input. Top-N (CA@N) will check whether any of the top-N translations generated by the model passes the test. Following TransCoder, we use the CA@1 metric computed with beam size 10.

**Experimental Details** Our model is a sequence-to-sequence (seq2seq) transformer model with 12 layers (6 in the encoder and 6 in the decoder), 8 attention heads, and a dimension of 1024. all programming language share a single encoder and a single decode, At training time, we alternate between each of programming language corpora. For multilingual translation language modeling task, we mask 15% of the token. In multilingual translation auto-encoding task, the implementation ratios of the three noise addition schemes, random token mask, random token dropout, and sentence permutation are 0.3, 0.3, and 0.2, respectively, and they are 0.5, 0.5, and 0 in the bag of words of distilled code. We optimize our model with the Adam optimizer and a polynomial decay learning rate scheduler, with an initial learning rate of 10-5 in most of our experiments. Our experiments use mixed precision accelerated training and are conducted in 8 V100 GPUs.

|  | Java→C# | | C#→Java | |
| Method | BLEU | CodeBLEU | BLEU | CodeBLEU |
|---|---|---|---|---|
| Naive copy | 18.54 | - | 18.69 | - |
| PLBART | 83.02 | 87.92 | 78.35 | 85.27 |
| StructCoder | 85.02 | 88.42 | 80.66 | 86.03 |
| CoDist | **85.80** | **88.75** | **83.19** | **86.20** |

Table 1: **Results on the CodeXGLUE translation task.** Our model achieves state-of-the-art performance on BLEU score of C#-Java and both BLEU and CodeBLEU on Java-C#.

## 5.2 Results

**Performance on CodeXGLUE** We conducted an evaluation of our proposed model on the CodeXGLUE benchmark, comparing it to two existing works, PLBART and StructCoder. PLBART, a strong baseline model, utilizes auto denoising pre-training tasks on Java and Python methods and natural language texts before fine-tuning on the task of program translation. StructCoder, on the other hand, employs abstract syntax trees and data flow graphs to make the encoder-decoder structure-aware, and currently holds the top position. The "naive copy" baseline simply replicates the input source code as the output translation, illustrating how closely similar two programming languages are. Table 1 presents our model's results which indicate its state-of-the-art performance in terms of BLEU score and CodeBLEU for both Java-C# and C#-Java translation tasks. This showcases the exceptional alignment capabilities of our method at the n-gram level.

**Performance on GeeksforGeeks** The experimental setup of TransCoder was followed, using the CA@1 metric calculated with a beam size of 10, for comparison of our model with TransCoder, TransCoder-ST, TransCoder-IR on Python-Java-C++ program translation pairs. Our model proved to have excellent results across various sub-translation tasks, as shown in Table 2, particularly in regards to starting from Python code. This success is mainly attributed to the distilled code design style which is more geared towards C++ and Java code, reducing the difficulty in aligning the target code from the Python code for the decompiler model. In contrast, TransCoder-ST requires the construction of a more precise automated unit-testing system to eliminate invalid translations, whereas our model does not require unit-testing and can be easily expanded to include more programming languages and other program translation models. We also compare with large language models InstructGPT (text-davinci-003

| Method | C++→Java | C++→Python | Java→C++ | Java→Python | Python→C++ | Python→Java |
|---|---|---|---|---|---|---|
| TransCoder | 65.1% | 47.1% | 79.8% | 49.0% | 32.6% | 36.6% |
| TransCoder-ST | 68.0% | 61.3% | 84.6% | 68.9% | 56.7% | 58.2% |
| TransCoder-IR | 62.9% | - | 74.5% | - | - | - |
| text-davinci-003 | 77.8% | 74.7% | 72.1% | 79.8% | 74.1% | 69.6% |
| gpt-3.5-turbo | **88.6%** | **85.2%** | 80.6% | **88.7%** | 85.67% | 80.9% |
| CoDist | 82.1% | 67.9% | **87.9%** | 68.1% | **86.9%** | **81.1%** |

Table 2: **Results on the TransCoder GeeksforGeeks translation task.** We follow the experimental configuration of TransCoder and ensure that all ground-truth test files in Python, Java and C++ output correct results in advance.

| Method | C++→Java | C++→Python | Java→C++ | Java→Python | Python→C++ | Python→Java |
|---|---|---|---|---|---|---|
| Codist | **82.14%** | **67.89%** | **87.85%** | **68.10%** | **86.94%** | **81.12%** |
| w/o MPG | 33.61% | 22.62% | 45.61% | 23.92% | 42.61% | 30.71% |
| w/o MLM | 70.12% | 61.21% | 80.08% | 60.78% | 77.94% | 71.36% |
| w/o MLM and DAE | 69.08% | 58.41% | 79.44% | 57.11% | 76.44% | 68.46% |

Table 3: **Abalation study of pretraining task in TransCoder GeeksforGeeks.** The setting of w/o MPG indicates that our model removes the multilingual program generation task, while w/o MLM and DAE means neither the masked language model task nor the denoising auto-encoding task are used. The rest of the configuration can be deduced by analogy.

version) (Ouyang et al., 2022) and ChatGPT (gpt-3.5-turbo version) (OpenAI., 2022) using the OpenAI official recommended translation prompt template, and turn off sampling to generate the final code. We cannot be sure of the generalization of OpenAI models as their test sets are proprietary, so we cannot rule out the influence of data leakage. While our model size is significant smaller than those models. ChatGPT potentially has state of the art translation rates for half the language pairs in GeeksForGeeks, but even still CoDist has the best performance on the other half of langauge pairs.

## 5.3 Analysis

We utilize a model that incorporates pre-training tasks (MLM, DAE and MPG) to enhance the performance of our distilled code decompiler. To examine the contribution of each pre-training task, we select the TransCoder GeeksforGeeks dataset and conduct experiments by removing some of the tasks. As demonstrated in Table 3, the results show that MLM and DAE tasks help model better understand semantics of distilled code and programming languages, while MPG task form and downstream translation task form are unified, which narrows the gap between the pre-training and finetune stages, and significantly improves the effect of the model. Our research shows that a judicious combination of these pre-training tasks can lead to more advanced and efficient code decompile.

## 6 Conclusion

In this paper, we propose a novel method for program translation with distilled code. Our distilled code has the characteristics of high information density and general applicability, making it an ideal candidate for serving as a translation pivot. We have developed distilled code compilers for C#, Java, Python, and C++, and a new multilingual program translation model. This model unifies multiple translation pairs into the distilled code decompiler task. Our results demonstrate the competitiveness of the proposed approach on the CodeXGLUE and GeeksforGeeks datasets, and highlights the potential for building good translation pivots in program translation tasks. We argue that our distillation idea has universality and is a highly efficient information compression method for structured texts like high-level programming languages, documents and tables with minimal data loss.

## Limitations

LLMs frequently encounter the issue of hallucinations, where generated text invents facts, objects or relationships. Our approach is not devoid of this problem, especially when method invocation has a complex web of dependencies. The insufficient availability of factual knowledge and world knowledge in the training corpus is primarily responsible for the current limitations of our model. In future work we will incorporate a Retrieval-Enhanced mechanism to obtain missing information from extensive databases in order to enhance the model.

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

# A Code Representation Analysis

Code developers often use descriptive and informative identifiers when writing programs to enhance the code's readability and maintainability. This practice of using informative identifiers helps to maintain the rich semantics of the code. In this section, we aim to investigate the influence of various code components(identifiers, program reserved words, token order etc.) on the semantic understanding. The findings of this analysis guides us in designing of distilled code.

Our research endeavors to identify which nodes of the abstract syntax tree can be removed without compromising the model's ability to comprehend the code. In the context of code pre-training models, our experiments aim to investigate the impact of adding noise to a specific code part. If the performance of these models significantly decreases as a result, it provides evidence that the semantics implied in the code part have a substantial influence on the model's ability to comprehend the code. The worse the performance of the model with noisy code parts, the more likely these parts damaged by noise contains core information and should be kept in our distilled code.

Specifically, we inject noise into some code parts by replacing the identifiers in code, which is proposed in DOBF (Roziere et al., 2021a), shuffling the original code , which is actually represented by the positional embeddings fed to the models, deleting the reserved keywords (e.g. 'assert', 'const', and 'if') and deleting the structure related symbols including brackets or certain punctuation symbols (i.e. '(', ')', '[', ']', '', '', ',', '.', and ';'). We select Java and C# as our target programming languages and use the corresponding datasets from CodeNet (Puri et al., 2021). Noticing that problems in CodeNet with larger indexes tend to have less valid solutions, we filter the dataset by selecting the first 1000 problems and randomly sampling 10 solutions for each problem, and also filter out the problems that are too short which are possibly wrong answers and remove the comments and the documents in the codes to avoid influences from natural languages in the codes. Finally the filtered dataset is composed of 7454 Java solutions and 4508 C# solutions. We obfuscate the identifiers to Java codes with tools borrowed from (Roziere et al., 2021a). Each identifier is replaced with a corresponding token like 'FUNC_0' and 'VAR_0'. After such changes, the code examples still pre-

```
/* original*/
public int[] twoSum(int[] nums, int target) {
  Map<Integer, Integer> dic = new HashMap<>();
  for (int i = 0; i < nums.length; ++i) {
    if (dic.containsKey(nums[i]))
      return new int[] {dic.get(nums[i]), i};
    else
      dic.put(target - nums[i], i);
  }
  return null;
}

/* obfuscated*/
public int[] FUNC_0(int[] VAR_0, int VAR_1) {
  Map<Integer, Integer> VAR_2 = new HashMap<>();
  for ( int VAR_3 = 0 ; VAR_3 < VAR_0.length; ++VAR_3) {
    if (VAR_2.containsKey(VAR_0[VAR_3]))
      return new int [] {VAR_2.get(VAR_0[VAR_3]), VAR_3};
    else
      VAR_2.put(VAR_1 - VAR_0[VAR_3], VAR_3);
  }
  return null;
}
```

Figure 3: **Full obfuscation version of the two-sum solution.** Full obfuscation of a two-sum solution which finds the two numbers in a list that will sum up to a target number in Java. The original code is shown above the obfuscated code.

```
/* shuffle lines*/
  Map<Integer, Integer> dic = new HashMap<>();
}
  for (int i = 0; i < nums.length; ++i) {
    else
      dic.put(target - nums[i], i);
      return new int[] {dic.get(nums[i]), i};
  return null;
    if (dic.containsKey(nums[i]))
  }
public int[] twoSum(int[] nums, int target) {

/* shuffle tokens*/
] i nums, ( . dic put ) int > ( ) null return { , int ] ( ]
 ] ; dic nums new = return int for ] } containsKey ; ]
twoSum i ) nums target { ) HashMap ] i ) ] . if ( Map get =
i . - ( i ] Integer, int ++i ; i } nums new dic int else
nums dic length target ( ) < ] Integer ; > i ) ( public ; 0
 , . < { ; < ] } ]
```

Figure 4: **Shuffled version of the two-sum solution.** The code with all lines shuffled is shown above the code with all tokens shuffled.

serve the functionality of the original codes. We show some examples of the obfuscated code, shuffled code and deleted code are shown separately in Figure 3, Figure 4 and Figure 5.

We expect some probe tasks to evaluate whether a well-behaved code representation model is more confused about the noised code compare with the original code. We hypothesize that if the specific code part is damaged and the model's performance drops significantly on these tasks, it indicates that the model relies on this piece of semantic information to understand the overall code.

The first probe task is the zero-shot code-to-code search task proposed in UniXcoder (Guo et al., 2022), which retrieve codes with the same semantics from a collection of candidates in a zero-shot setting and rank the candidates with these similarities. Denote the obfuscation transform from code space $\mathcal{W}$ to $\mathcal{W}_{obfuscated}$ by $f_{obf}$. In practice, we

```
/* remove keywords*/
[ ] twoSum  ( [ ] nums , target ) {
  Map < Integer , Integer > dic = HashMap < > ( ) ;
  ( i = 0 ; i < nums . length ; ++ i ) {
    ( dic . containsKey ( nums [ i ] ) )
      [ ] { dic . get ( nums [ i ] ) , i } ;
      dic . put ( target - nums [ i ] , i ) ;
  }
  ;
}

/* remove certain symbols*/
public int twoSum int nums int target
  Map < Integer Integer > dic = new HashMap < >
  for int i = 0 i < nums length ++ i
    if dic containsKey nums i
      return new int dic get nums i i
    else
      dic put target - nums i i
  return null
```

Figure 5: **Deleted keywords Version of the two-sum solution.** The code with keywords removed is shown above the code with structural symbols replaced with blank spaces.

can use the original code $w_{i,j}$ as query and the obfuscated codes $f_{obf}(w_{i,j})$ as candidates, and vice versa. For an input code $w_{i,j}$ which is the $j^{th}$ solution to the $i^{th}$ problem, we consider all solutions for the same problem as the correct retrieval results, i.e. $w_{i,j'}$ and $f_{obf}(w_{i,j'})$, and all the solutions for other problems as incorrect retrieval results, i.e. $w_{i',j}$ and $f_{obf}(w_{i',j})$ where $i' \neq i$.

| unixcoder-base | | | | |
|---|---|---|---|---|
| qurey-cand | bleu@1 | p@10 | map | mrr |
| Java-Java | 53.49 | 23.81 | 28.78 | 36.03 |
| Java-Java(shuffle) | 73.58 | 22.5 | 26.88 | 41.97 |
| Java-Java(keywords) | 83.09 | 24.3 | 29.49 | 40.21 |
| Java-Java(symbols) | 19.9 | 21.91 | 26.38 | 40.38 |
| Java-Java(dobf) | **2.59** | **8.43** | **9.62** | **18.23** |

Table 4: **Compare code-code search task performance appling various noise addition methods.** Java-Java means the ability of the Unix-Coder base to search for all Java codes with the same semantics from the candidate codes using a certain Java code as a query. Java(shuffle), Java(keywords), Java(symbols) and Java(dobf) represents the noise of adding shuffle code, delete the reserved keywords, delete the structure related symbols and replace the identifiers to all the codes in the candidate library.

Table 4 shows the interference of four types of noise on the zero-shot code-to-code search task. The BLEU@1 is the BLEU score between the query and the candidate with the highest similarity. The P@10 is the percentage of the queries that the correct code is in the top 10 most similar codes. The MAP is the average of the average precision of the correct code for each query. The MRR is the average of the reciprocal rank of the correct code for each query. These four metrics comprehensively measure the quality of the retrieved results. Compared with the baseline, shuffling the original code and deleting reserved keywords does not interfere with the model's performance, which means the

model can tolerate the noise we add when designing the translation pivot to flur different high-level languages. Meanwhile replacing identifiers and deleting structure-related symbols significantly reduces the model's effectiveness. This indicates that identifiers and structure-related symbols are core information that should be kept in the translation pivot as much as possible.

Another probe task falls on the semantic analysis. Here our study aims to analyze the code semantic understanding of the model by analysing the distribution of cosine similarity on positive and negative code pairs, where each positive and negative examples serve as the query and the other positive and negative examples in the candidate set serve as the candidate. We collect all the positive code pairs that are composed of any two solutions to the same question, and the negative pairs is composed of any two solutions to different questions. In the experiment, we inject noise into the positive and negative examples and observe the changes of their density plots of cosine similarities. We should try to ensure that adding noise to the model will not prevent it from distinguishing between positive and negative.

Denote the original code of the $j^{th}$ solution to the $i^{th}$ problem by $w_{i,j}$ and its embedding from a pre-trained model by $\mathbf{h}_{i,j}$. We apply average pooling to the outputs from the last layer of the pre-trained model to get the embedding $\mathbf{h}_{i,j}$ of the input code $w_{i,j} \in \mathcal{W}$. All the positive code pairs and the negative pairs can be denoted as $\mathcal{P} = \{(w_{i,j}, w_{i,k})\}_{j \neq k}$, and $\mathcal{N} = \{(w_{i,j}, w_{i',k})\}_{i \neq i'}$ respectively. After computing the cosine similarities of the embeddings of the code pairs in $\mathcal{P}$ and $\mathcal{N}$, we plot the histograms of the similarities to observe the changes in the similarity distribution of positive and negative under different noise additions in Figure 6.

Figure 6 shows the similarity distribution of positive and negative code pairs under different noise additions. The green histogram and yellow histogram in each subfigure represents the similarity score density distribution of positive pairs and negative pairs. Ideally, the similarity distribution of negative examples should tend to 0 as a whole, while the similarity distribution of positive examples should tend to 1 as a whole. Therefore, the larger the overlap between the yellow histogram and the green histogram, the more confused the code model is in distinguishing the semantic dif-

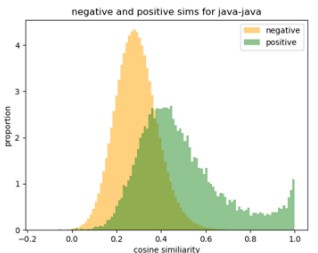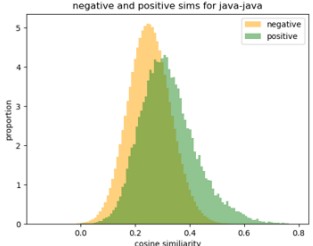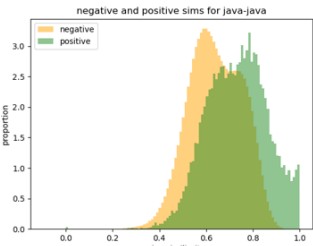

Figure 6: **Similarity distribution of positive and negative code pairs under different noise additions.** The three subfigures in our study represent Java2Java (no noise added), Java2Java-dobf (dobf noise added to candidate set), and Java-dobf2Java-dobf (dobf noise added to both query and candidate) from left to right. The yellow histogram in each subfigure represents the cosine similarity distribution between each negative example and the rest of the negative examples. We calculated the proportion of code pairs occupying different similarities and formed the similarity score density distribution of the overall negative example pairs. The green histogram in each subfigure represents the similarity score density distribution of positive pairs.

ference between positive and negative examples. To analyze this, we introduced noise such as replacing the identifiers in code (dobf), and gradually increased this noise in the query and candidate parts in the sub-images from left to right. Our results showed that as more dobf noise was added to code pairs, the similarity overlap between positive and negative code pairs increased, and the overall similarity distribution gradually moved closer to 1. This suggests that the semantic information in the identifiers part is important for the model to correctly understand and distinguish codes with different semantics.

## B  Unify Basic Morphemes

Here we demonstrate part of morphemes with equivalent semantics across the Python, Java, C++, and C# programming languages, and present these morphemes in their unified expression forms. To aid in models pretrained on common code corpora capable of comprehending the semantics of these morphemes, we often adopt their unified representations from the symbols and reserved words found in various high-level programming languages.

| Unified Form | C++ | Java | C# | Python |
|---|---|---|---|---|
| int a | int a | int a | int a | int a |
| float a | float a | float a | float a | float a |
| string a | std::string a | String a | string a | str a |
| bool a | bool a | boolean a | bool a | bool a |
| char a | char a | char a | char a | - |
| vector<> a | vector<> a | Vector<> a | List<> a | a=[] |
| map<> a | std::map<> a | HashMap<> a | Dictionary<> a | a={} |
| set<> a | std::set<> a | HashSet<> a | HashSet<> a | a=set() |
| queue<> a | std::queue<> a | Queue<> a | Queue<> a | a=queue.Queue() |
| deque<> a | std::deque<> a | Deque<> a | - | a=deque() |

Table 6: **Unify part of common data types.**

| Unified Form | C++ | Java | C# | Python |
|---|---|---|---|---|
| sqrt(a) | sqrt(a) | Math.sqrt(a) | Math.Sqrt(a) | math.sqrt(a) |
| log(a) | log(a) | Math.log(a) | Math.Log(a) | math.log(a) |
| floor(a) | floor(a) | Math.floor(a) | Math.Floor(a) | math.floor(a) |
| rand(a,b) | rand(b-a)%+b | rand.nextInt(b-a)+b | rand.Next(a,b) | random.randint(a,b) |
| print(a) | cout«a | System.out.print(a) | Console.Write(a) | print(a, end='') |
| println(a) | count«a«endl | System.out.println(a) | Console.WriteLine(a) | print(a) |
| islower(a) | islower(a) | Character.isLowerCase(a) | Char.IsLower(a) | a.islower() |
| tolower(a) | tolower(a) | Character.toLowerCase(a) | Char.ToLower(a) | a.tolower() |
| replace(c,a,b) | c.replace(a,b) | c.replace(a,b) | c.replace(a,b) | c.replace(a,b) |
| length(a) | a.length() | a.length() | a.Length | len(a) |

Table 7: **Unify part of common built-in methods.**

Table 5, Table 6 and Table 7 are presented the unified forms of operators, data types, and built-in methods partly. Here we strive to align the vast majority of common basic morphemes in C++, Java, C#, and Python language. For those that we do not yet support, alignment will be handled by the fuzzy strategy in the compiler and the multilingual code snippet generation task in the decompiler.

| Unified Form | C++ | Java | C# | Python |
|---|---|---|---|---|
| a+b | a+b | a+b | a+b | a+b |
| a-b | a-b | a-b | a-b | a-b |
| a*b | a*b | a*b | a*b | a*b |
| a/b | a/b | a/b | a/b | a/b |
| a%b | a%b | a%b | a%b | a%b |
| int(a/b) | int(a/b) | int(a/b) | int(a/b) | a//b |
| pow(a,b) | pow(a,b) | Math.pow(a,b) | Math.pow(a,b) | a**b |
| a&&b | a&&b | a&&b | a&&b | a and b |
| a\|\|b | a\|\|b | a\|\|b | a\|\|b | a or b |
| !a | !a | !a | !a | not a |

Table 5: **Unify part of common operators.**