# OpenReview forum: "Program Translation via Code Distillation"
_EMNLP/2023/Conference — EMNLP 2023 Main_

### Official Review · Reviewer_soRb · 2023-08-04

**Typos Grammar Style And Presentation Improvements:** N/A
**Soundness:** 3

**Excitement:**

4: Strong: This paper deepens the understanding of some phenomenon or lowers the barriers to an existing research direction.

**Missing References:**

N/A

**Paper Topic And Main Contributions:**

The paper introduces a novel model named Code Distillation (CoDist) for code translation. This model effectively captures the semantic and structural equivalence of code in a language-agnostic intermediate representation, enabling it to serve as a translation pivot for any programming language. Experimental results demonstrate that the proposed model outperforms the baselines on both the CodeXGLUE and GeeksForGeeks translation benchmarks. The main contributions of this research include: (1) the introduction of a new programming translation pivot language called distilled code, and (2) the utilization of multiple unsupervised code corpora through three distinct self-supervised pre-training tasks.

**Questions For The Authors:**

Why is there no experiment conducted on w/o DAE in Table 3?



**Reasons To Accept:**

1. This paper introduces a novel pivot language for code translation and investigates three distinct self-supervised pre-training tasks.
2. The motivation behind the study is lucid, and the paper is thoughtfully composed and well-written.

**Reasons To Reject:**

1. The paper lacks sufficient ablation analysis concerning the distilled code methods. For instance, it does not explore the impact of excluding syntax tree pruning, unify basic morphemes, or fuzzing remaining variations individually.
2. The unify basic morphemes approach necessitates numerous manual operations, making it challenging to extend its applicability to additional languages.

**Reproducibility:**

3: Could reproduce the results with some difficulty. The settings of parameters are underspecified or subjectively determined; the training/evaluation data are not widely available.

**Reviewer Confidence:**

3: Pretty sure, but there's a chance I missed something. Although I have a good feel for this area in general, I did not carefully check the paper's details, e.g., the math, experimental design, or novelty.

---

> ### Author Rebuttal · Authors · 2023-08-28
>
> Thank you time and energy to review our manuscript! We read your opinions carefully and we want to explain further about these points of opinions.
>
> 1. The paper lacks sufficient ablation analysis concerning the distilled code methods. For instance, it does not explore the impact of excluding syntax tree pruning, unify basic morphemes, or fuzzing remaining variations individually.
>
> We have added some experiments here for your reference.
>
> 1.1. Analyze the role of the three steps in building a translation pivot
>
> |distilled code type  |BLEU|CodeBLEU|
> |------|------|------|
> |tree pruning|74|68|
> |tree pruning + fuzzing|76|71|
> |All(tree pruning + fuzzing+unify)|81|78|
>
> We take the pair code of C++ and Java in the Transcoder geeksforgeeks evaluation data, and we use three strategies to extract their distiiled code (only syntax tree pruning, syntax tree pruning + fuzziness, and all strategies are used), and calculate the difference between the distilled code of the pair pair code between them. We measured their "distance" using BLEU and CodeBLEU, and the results are shown in the table above, and we can see that the addition of these three strategies can gradually bridge the differences between different language codes and bring them closer like a perfect translation pivot.
>
> 1.2 2.Analyze the role of the three steps in finall translation task
>
> Computation Accuracy on TransCoder GeeksforGeeks dataset:
>
> |distilled code type  |C++2Java|C++2Python|Java2C++	|Java2Python|Python2C++|Python2Java|
> |------|------|------|------|------|------|------|
> |tree pruning|63.90%|44.61%|74.95%|50.86%|51.82%|52.90%|
> |tree pruning + fuzzing|68.46%|50.43%|76.02%|55.30%|57.68%|60.17%|
> |All(tree pruning + fuzzing+unify)|82.14%|67.89%|87.85%|68.10%|86.94%|81.12%|
> | Compared model|||||
> |TransCoder|65.1%|47.1%|79.8%|49.0%|32.6%|36.6%|
>
> BLEU & CodeBLEU on CodeXGLUE dataset
>
> |distilled code type|C#2Java BLEU|C#2Java CodeBLEU|Java2C# BLEU|Java2C# CodeBLEU|
> |------|------|------|------|------|
> |tree pruning|75.26|79.52|77.98|82.26|
> |tree pruning + fuzzing|83.01|82.98|83.14|86.58|
> |All(tree pruning + fuzzing+unify)|83.19|86.12|85.8|88.75|
> | Compared model|||
> |CodeBERT|72.14|79.41|79.92|85.10|
>
> We can observe that the degree of contribution of these three parts is different in different types of code translation datasets. The Transcoder Geeksforgeeks dataset comes from programming test websites, and its titles rarely involve complex external API calls, so fuzzing remaining variations such as on the type like method invocation are relatively rare; The code in CodexGLUE comes from the industry's github repro, so there are a large number of cross-function and even cross-folder level API calls, so our fuzzy method can significantly improve the effect.
>
> 2. The unify basic morphemes approach necessitates numerous manual operations, making it challenging to extend its applicability to additional languages.
>
> The appendix B of the paper and our feedback here can be used for your reference. Our unified basic themes here mainly include operators, primitive data types, and build-in methods parts. The first two parts can be easily enumerated and their concepts are commonly shared by most programming languages. The build-in methods are also enumerated and shared across all popular languages (e.g., print, sort, etc.)
>
> In the codist method, for non enumerable code, we actually rely on fuzzy part of code and parsing tree prune to complete intermediate language extraction, which completely avoids manual components and can automatically generate samples without human intervention.
>
> In short, Codist is designed to be an unsupervised algorithm. Its main body alignment is achieved through syntax tree pruning and fuzzification operations. Manual alignment is limited and enumerable, yet can reduce the noise significantly and relieve the neural networking understanding burdens.
>
> 3. Why is there no experiment conducted on w/o DAE in Table 3?
>
> We have added some experiments here for your reference.
>
> |pretrain method  |C++2Java|C++2Python|Java2C++	|Java2Python|Python2C++|Python2Java|
> |------|------|------|------|------|------|------|
> |Codist|82.14%|67.89%|87.85%|68.10%|86.94%|81.12%|
> |w/o DAE|71.18%|64.01%|80.94%|64.22%|79.44%|70.54%|

---

### Official Review · Reviewer_KDdW · 2023-08-05

**Soundness:** 4

**Excitement:**

4: Strong: This paper deepens the understanding of some phenomenon or lowers the barriers to an existing research direction.

**Paper Topic And Main Contributions:**

The authors present a novel approach to code translation using intermediate code representation. The main idea is to use a synthetic language code as intermediate representation. The main contribution are the method of intermediate code representation and a model for the translation using this representation.

**Reasons To Accept:**

The authors in this paper are reached the holy grail for NLP in the different domain, the grail I mentioned is interlingua, a universal language. The authors developed interlingua for the programming languages and demonstrated its usefulness.

**Reasons To Reject:**

The invented language is limited to single object-oriented paradigm in programming, while there are other programming languages following different paradigms.

**Reproducibility:**

4: Could mostly reproduce the results, but there may be some variation because of sample variance or minor variations in their interpretation of the protocol or method.

**Reviewer Confidence:**

4: Quite sure. I tried to check the important points carefully. It's unlikely, though conceivable, that I missed something that should affect my ratings.

---

> ### Author Rebuttal · Authors · 2023-08-28
>
> Thank you time and energy to review our manuscript! We read your opinions carefully and we want to explain further about these points of opinions.
>
> 1. The invented language is limited to single object-oriented paradigm in programming, while there are other programming languages following different paradigms.
>
> In fact, our code distillation method is not limited to object-oriented programming languages. In theory, all of the programming languages which can be parsing into syntax trees can be implemented through our method. we use Treesitter (supporting 113 program languages) as parsing tools.
>
> The current limitation to our approach is rooted in the available evaluation datasets within this code translation domain. Regrettably, the current datasets (TransCoder GeeksforGeeks and CodeXGLUE)  does not encompass all programming language types. This is the main reason we presently apply this approach only to object-oriented languages. This is just a coincidence and not a limitation of our code distillation method. It's easy to extend this approach to other classes of languages, such as procedural-oriented languages.  In order to better understanding, we have given the following examples:
>
> 1.1 Process-oriented code example
>
> C implementation of bubble sort:
> ```
> void swap(int arr[], int i, int j) {
>     int temp = arr[i];
>     arr[i] = arr[j];
>     arr[j] = temp;
> }
>
> void bubbleSort(int arr[], int size) {
>     for (int i = 0; i < size - 1; i++) {
>         for (int j = 0; j < size - i - 1; j++) {
>             if (arr[j] > arr[j + 1]) {
>                 swap(arr, j, j + 1);
>             }
>         }
>     }
> }
> ```
> According to CoDist's compilation rules, the distilled code of this code should be
> ```
> swap(arr, i, j){
>     temp = arr[i];
>     arr[i] = arr[j];
>     arr[j] = temp;
> }
>
> bubbleSort(arr, size){
>     for(i=0; i<size-1; i=i+1){
>         for(j=0; j<size-i-1; j=j+1){
>             if(arr[j]>arr[j+1]){
>                 BOW: swap ( arr , j , j + 1 )
>             }
>         }
>     }
> }
> ```
>
> 1.2. Functional programming code example
>
> Scheme  implementation of bubble sort:
> ```
> (define (swap arr i j)
>   (let ((temp (vector-ref arr i)))
>     (vector-set! arr i (vector-ref arr j))
>     (vector-set! arr j temp)))
>
> (define (bubble-sort arr size)
>   (do ((i 0 (+ i 1)))
>       ((< i (- size 1)))
>     (do ((j 0 (+ j 1)))
>         ((< j (- size i 1)))
>       (if (> (vector-ref arr j) (vector-ref arr (+ j 1)))
>           (swap arr j (+ j 1)))))
>   arr)
> ```
> In Scheme, prefix notation is used to represent computational processes. Here at Distilled Code, we will uniformly convert it into infix notation to align with other languages. Then according to CoDist's compilation rules, the distilled code of this code should be：
> ```
> swap(arr, i, j){
>     temp = arr[i]
>     arr[i] = arr[j]
>     arr[j] = temp
> }
>
> bubble-sort(arr, size){
>     for(i=0; i<size-1; i=i+1){
>         for(j=0; j<size-i-1; j=j+1){
>             if(arr[j]>arr[j+1){
>                 BOW: swap ( arr , j , j + 1 );
>             }
>         }
>     }
>     return arr;
> }
> ```
> 1.3. Object-oriented code example
>
> Python implementation of bubble sort:
> ```
> def swap(arr, i, j):
>     temp = arr[i]
>     arr[i] = arr[j]
>     arr[j] = temp
>
> def bubble_sort(arr):
>     for i in range(len(arr) - 1):
>         for j in range(len(arr) - i - 1):
>             if arr[j] > arr[j + 1]:
>                 swap(arr, j, j + 1)
> ```
> According to CoDist's compilation rules, the distilled code of this code should be
> ```
> swap(arr, i, j){
>     temp = arr[i];
>     arr[i] = arr[j];
>     arr[j] = temp;
> }
>
> bubbleSort(arr){
>     for(i=0; i<len(arr)-1; i=i+1){
>         for(j=0; j<len(arr)-i-1; j=j+1){
>             if(arr[j]>arr[j+1]){
>                 BOW: swap ( arr , j , j + 1 )
>             }
>         }
>     }
> }
> ```
> Here, we've chosen three different programming paradigms for demonstration: procedural (C), functional (Scheme), and object-oriented (Python). We implemented the bubble sort algorithm in these languages and converted them to our Distilled Code. Despite the significant syntax differences among all languages, our approach can provide a highly unified representation. This demonstrates our method's ability to generalize across different programming styles.
>
> Thanks again for your perspective. This brings inspiration to our future work, and we may expand our method to other types of languages ​​in the future work(datset building and evaluation).

---

### Official Review · Reviewer_U8tt · 2023-08-06

**Soundness:** 3

**Excitement:**

4: Strong: This paper deepens the understanding of some phenomenon or lowers the barriers to an existing research direction.

**Paper Topic And Main Contributions:**

To overcome the the noise in code snippet  alignment and the diversity of IRs for code translation, the authors propose a novel model called  Code Distillation (CoDist) whereby they capture the semantic and structural equivalence of code in a language agnostic intermediate representation. Distilled code serves as a translation pivot for any programming language, leading by construction to parallel corpora which scale to all available source code by simply applying the distillation compiler. And the experiments show the proposed approach achieves state-of-the-art performance on CodeXGLUE and TransCoder GeeksForGeeks translation benchmarks.


**Reasons To Accept:**

- I like the idea of improving code translation by utilizing a language-agnostic IR as the translation pivot. I think creating the proposed IR that  captures the semantic and structural equivalence of code is very interesting and useful.
- All implementation details are presented clearly and resonably. The proposed method looks very convincing and solid.
- The proposed method achieved very good performance, and the main SOTA methos are included in baselines. It’s surprising that the model’s performance is close to GPT-3.5-turbo.


**Reasons To Reject:**

- Some ablation studies analizes should be conducted to verify the effectiveness of the components of the method.Therefore, the description of the method should be simplified.

**Reproducibility:**

3: Could reproduce the results with some difficulty. The settings of parameters are underspecified or subjectively determined; the training/evaluation data are not widely available.

**Reviewer Confidence:**

2: Willing to defend my evaluation, but it is fairly likely that I missed some details, didn't understand some central points, or can't be sure about the novelty of the work.

---

> ### Author Rebuttal · Authors · 2023-08-28
>
> Thank you time and energy to review our manuscript! We read your opinions carefully and we want to explain further about these points of opinions.
>
> 1. Some ablation studies analizes should be conducted to verify the effectiveness of the components of the method.Therefore, the description of the method should be simplified.
>
> Sorry, As we consider that code translation tasks are relatively minor in EMNLP submissions, and our method introduce some complex concepts. Therefore, we would like to elaborate in detail and thoughtfulness to reduce the difficulty of understanding for non code translation reviewers. Taking into account the opinions of the other two reviewers, we have added the following ablation experimental results, please refer to the experiments we added. We will also supplement these experiments in the official version.
>
> 1.1. Analyze the role of the code distillation steps(syntax tree pruning, unify basic morphemes, fuzzing remaining variations) in building a translation pivot
>
> |distilled code type  |BLEU|CodeBLEU|
> |------|------|------|
> |tree pruning|74|68|
> |tree pruning + fuzzing|76|71|
> |All(tree pruning + fuzzing+unify)|81|78|
>
> We take the pair code of C++ and Java in the Transcoder geeksforgeeks evaluation data, and we use three strategies to extract their distiiled code (only syntax tree pruning, syntax tree pruning + fuzziness, and all strategies are used), and calculate the difference between the pair of distilled code. We measured their "distance" using BLEU and CodeBLEU, and the results are shown in the table above, and we can see that the addition of these three strategies can gradually bridge the differences between different language codes and bring them closer like a perfect translation pivot.
>
> 1.2 2.Analyze the role of the code distillation steps(syntax tree pruning, unify basic morphemes, fuzzing remaining variations) in finall translation task
>
> Computation Accuracy on TransCoder GeeksforGeeks dataset:
>
> |distilled code type  |C++2Java|C++2Python|Java2C++	|Java2Python|Python2C++|Python2Java|
> |------|------|------|------|------|------|------|
> |tree pruning|63.90%|44.61%|74.95%|50.86%|51.82%|52.90%|
> |tree pruning + fuzzing|68.46%|50.43%|76.02%|55.30%|57.68%|60.17%|
> |All(tree pruning + fuzzing+unify)|82.14%|67.89%|87.85%|68.10%|86.94%|81.12%|
> | Compared model|||||
> |TransCoder|65.1%|47.1%|79.8%|49.0%|32.6%|36.6%|
>
> BLEU & CodeBLEU on CodeXGLUE dataset
>
> |distilled code type|C#2Java BLEU|C#2Java CodeBLEU|Java2C# BLEU|Java2C# CodeBLEU|
> |------|------|------|------|------|
> |tree pruning|75.26|79.52|77.98|82.26|
> |tree pruning + fuzzing|83.01|82.98|83.14|86.58|
> |All(tree pruning + fuzzing+unify)|83.19|86.12|85.8|88.75|
> | Compared model|||
> |CodeBERT|72.14|79.41|79.92|85.10|
>
> We can observe that the degree of contribution of these three parts is different in different types of code translation datasets. The Transcoder Geeksforgeeks dataset comes from programming test websites, the code of Geeksforgeeks most leverage on primative data types and expressions, so unify play the most important part of the translation; The code in CodexGLUE comes from the industry's github repro. There are a large number of cross-function and even cross-folder level API calls, so our fuzzy method can significantly improve the effect.
>
> 1.2.3 add more pretrain task abalation study
>
> |pretrain method  |C++2Java|C++2Python|Java2C++	|Java2Python|Python2C++|Python2Java|
> |------|------|------|------|------|------|------|
> |Codist|82.14%|67.89%|87.85%|68.10%|86.94%|81.12%|
> |w/o DAE|71.18%|64.01%|80.94%|64.22%|79.44%|70.54%|

---

### Meta-Review · Area_Chair_YZ3Z · 2023-09-14

**Recommendation:** 5

**Metareview:**

The paper addresses program translation and proposes an intermediate language (called distilled code) that prunes language-specific details but preserves the semantics of a program. (It's noticed that _distilling_ here has a different meaning from knowledge distilling).

Reviewers generally find the idea interesting - having an intermediate language hasn't been work really well for translating natural language but may be a good direction to go for translating programming language. Overall, the approach is well designed and achieves high performance.

---

### Decision · Program_Chairs · 2023-10-07

**Decision:**

Accept-Main

**Comment:**

The paper addresses program translation and proposes an intermediate language (called distilled code) that prunes language-specific details but preserves the semantics of a program. (It's noticed that _distilling_ here has a different meaning from knowledge distilling).

Reviewers generally find the idea interesting - having an intermediate language hasn't been work really well for translating natural language but may be a good direction to go for translating programming language. Overall, the approach is well designed and achieves high performance.